# Oxazepam and cognitive reappraisal: A randomised experiment

**Gustav Nilsonne**[1,2]*, **Sandra Tamm**[1,2,3], **Armita Golkar**[1,4], **Andreas Olsson**[1], **Karolina Sörman**[1], **Katarina Howner**[1], **Marianne Kristiansson**[1], **Martin Ingvar**[1], **Predrag Petrovic**[1]

**1** Department of Clinical Neuroscience, Karolinska Institutet, Stockholm, Sweden, **2** Department of Psychology, Stockholm University, Stockholm, Sweden, **3** Department of Psychology, University of Oxford, Oxford, England, **4** Department of Clinical Psychology, University of Amsterdam, Amsterdam, Netherlands

* gustav.nilsonne@ki.se

## Abstract

### Background

Cognitive reappraisal is a strategy for emotional regulation, important in the context of anxiety disorders. It is not known whether anxiolytic effects of benzodiazepines affect cognitive reappraisal.

### Aims

We aimed to investigate the effect of 25 mg oxazepam on cognitive reappraisal.

### Methods

In a preliminary investigation, 33 healthy male volunteers were randomised to oxazepam or placebo, and then underwent an experiment where they were asked to use cognitive reappraisal to upregulate or downregulate their emotional response to images with negative or neutral emotional valence. We recorded unpleasantness ratings, skin conductance, superciliary corrugator muscle activity, and heart rate. Participants completed rating scales measuring empathy (Interpersonal Reactivity Index, IRI), anxiety (State-Trait Anxiety Inventory, STAI), alexithymia (Toronto Alexithymia Scale-20, TAS-20), and psychopathy (Psychopathy Personality Inventory-Revised, PPI-R).

### Results

Upregulation to negative-valence images in the cognitive reappraisal task caused increased unpleasantness ratings, corrugator activity, and heart rate compared to downregulation. Upregulation to both negative- and neutral-valence images caused increased skin conductance responses. Oxazepam caused lower unpleasantness ratings to negative-valence stimuli, but did not interact with reappraisal instruction on any outcome. Self-rated trait empathy was associated with stronger responses to negative-valence stimuli, whereas self-rated psychopathic traits were associated with weaker responses to negative-valence stimuli.

**Data Availability Statement:** Data and analysis code for this paper are openly available at https://doi.org/10.5281/zenodo.3903120.

**Funding:** This work was supported by the Swedish Society for Medicine and the Osher Center for

Integrative Medicine. he funders had no role in study design, data collection and analysis, decision to publish, or preparation of the manuscript.

**Competing interests:** The authors have declared that no competing interests exist.

## Conclusions

While 25 mg oxazepam caused lower unpleasantness ratings in response to negative-valence images, we did not observe an effect of 25 mg oxazepam on cognitive reappraisal.

## Introduction

### Background

Emotional regulation is an important aspect of normal behavior in healthy individuals and often altered in patients with psychiatric disorders, including emotional instability and anxiety [1, 2], suggesting a less effective top-down control of emotional processes. Pharmacological substances such as benzodiazepines provide a rapid anxiolytic effect, but are associated with a risk for dependency. Anecdotally, benzodiazepines have been reported to be used to disinhibit criminal violent behavior [3–5]. This disinhibition theory of criminal violent behaviour suggests that either empathy processes or top-down regulatory processes are suppressed by benzodiazepines. However, previous studies have not tested whether benzodiazepines affect top-down regulation of emotion.

One strategy to regulate emotions is through reappraising emotional stimuli in a non-emotional way. Cognitive reappraisal represents an explicit top-down regulatory mechanism in the processing of emotional stimuli [6]. In functional brain imaging studies, cognitive reappraisal has been associated with activity particularly in dorsolateral prefrontal cortex (dlPFC) and lateral orbitofrontal cortex (OFC) [7–11]. These areas are proposed to exert top-down control over emotional processing in limbic brain structures including the amygdala [7–12]. Cognitive reappraisal is important in psychiatric conditions involving anxiety [13–15], which may be associated with insufficient top-down control [14, 16]. One meta-analysis of fMRI studies found that patients with mood and anxiety disorders showed less activation than healthy controls during cognitive reappraisal in areas including the bilateral dorsomedial prefrontal cortex and the left ventrolateral prefrontal cortex [17]. Training in cognitive reappraisal can also be part of a treatment for depression, as well as other psychiatric conditions (reviewed in [18])

Benzodiazepines are anxiolytic drugs acting through $GABA_A$ receptors, which are pentameric ligand-gated ion channels composed of $\alpha$, $\beta$, and $\gamma$ subunits. Anxiolytic effects of benzodiazepines are suggested to be mediated primarily by the $\alpha$-2 subunit containing $GABA_A$ receptors [19], expressed particularly in the amygdala [20]. Conversely, sedative and anticonvulsant effects are likely mediated mainly by $\alpha$-1 subunit containing $GABA_A$ receptors, highly expressed in the cerebral cortex [19–22]. In laboratory settings, benzodiazepines have been shown to enhance the response to positive vs negative words, modulate emotional memory, and inhibit recognition of facial expressions of anger [23, 24]. It has been shown that increased activity in amygdala and insula associated to fear and anxiety processing are suppressed in a dose-dependent manner by treatment with benzodiazepines in humans [12, 25, 26]. One benzodiazepine with a clear anxiolytic effect, but with less sedative properties than many other benzodiazepines is oxazepam. We have previously tested the hypothesis that 20 mg oxazepam would inhibit empathy for pain, finding no conclusive evidence for such an effect [27]. Another possible explanation for instrumental use of benzodiazepines could be that they, often in combination with alcohol, reduce the ability to regulate emotional responses and thereby cause increased aggression. This would suggest an interaction between top-down regulatory mechanisms and treatment with benzodiazepines. This hypothesis is especially interesting

since areas in prefrontal cortex that are involved in emotional regulation have high concentrations of $GABA_A$ receptors, suggesting a putative mechanism by which the $GABA_A$ system may impact emotional regulation efficiency [28].

Thus, both cognitive reappraisal and benzodiazepines may be associated with inhibition of unpleasantness in response to negatively valenced emotional stimuli, and both act upon the amygdala. However, it is not known whether these regulatory processes interact or act independently on emotional processing.

We investigated the effects of 25 mg oxazepam in a study encompassing three different experiments targeting different types of emotional processing: emotional mimicry, empathy for pain, and cognitive reappraisal. This paper reports results from the cognitive reappraisal experiment. Results from the experiments on emotional mimicry and empathy for pain have been previously reported [27], and the primary findings were that 25 mg oxazepam did not have a major effect on emotional mimicry nor on empathy for pain. Cognitive reappraisal can be thought of as a higher-level top-down emotion regulating function, mechanistically distinct from the more bottom-up processes of emotional mimicry and empathy for pain.

### Aims

We aimed to investigate effects of 25 mg oxazepam and cognitive reappraisal on emotion-related outcomes. The main hypothesis was that oxazepam would be associated with a reduced ability to regulate emotions through cognitive reappraisal. To capture subjective experience as well as psycho-physiological aspects of emotion, we investigated participants' ratings of unpleasantness, skin conductance, heart rate and facial EMG. Additionally, we explored associations of personality traits related to empathy, psychopathy and anxiety to emotional regulation.

## Materials and methods

The study was approved by the regional ethical review board of Stockholm (no. 2009/1128-31/3). Participants gave written informed consent.

### Study design

This experiment formed part of a larger study on the effects of oxazepam on social emotional processes. For a detailed description, see [27]. Briefly, participants were randomised to 25 mg oxazepam or placebo in a double-blind between-groups design, and underwent experiments on emotional mimicry, empathy for pain, and cognitive reappraisal. This paper reports effects of oxazepam on cognitive reappraisal. The study was performed in two waves. Stimuli were balanced over regulation instructions using two different trial lists. Due to an error in randomisation in wave 1, stimulus images were not balanced between instructions to upregulate and downregulate emotional response. Therefore only data from wave 2 were analysed for the experiment on cognitive reappraisal.

### Participants

As described in [27], participants were required to be right-handed, male, 18-45 years of age, to have no history of neurological or psychiatric disease including substance abuse, to speak and understand Swedish fluently, and not to be habitual consumers of nicotine, to reduce the risk of abstinence symptoms during the experiment. Furthermore, students of psychology, behavioural sciences, and medicine (past the 3rd semester) were not included, because training in medicine may cause a more detached attitude towards images of injured and sick people,

which were used in the experiment, and because students of psychology and behavioural science may be prone to metacogitate and use different strategies for emotional regulation. We aimed for a sample size of $n = 40$, with 20 participants in each of the two treatment groups, based on pragmatic considerations. Participants were paid 500 SEK (approx. 50 Euro or 60 USD), subject to tax.

## Stimuli and experimental paradigm

The experimental paradigm was adapted from [7]. Participants were shown negative and neutral stimuli following an instruction to either upregulate or downregulate their emotional response, see Fig 1. The reappraisal instruction was shown for 2 seconds, followed immediately by the image, which was shown for 1 second. Stimulus images were chosen from the International Affective Picture System (IAPS) [29]. On normative ratings, negative images had a mean valence of 20.2 [SD 0.25], and neutral images had a mean valence of 5.02 [0.05]. Participants were asked to either upregulate or downregulate their emotional response by cognitive reappraisal, i.e. imagining different contexts for the situations shown in the images, such as a fictitious situation (downregulation) or something happening to someone close to them (upregulation). Participants were specifically instructed not to close their eyes or look away. In total,

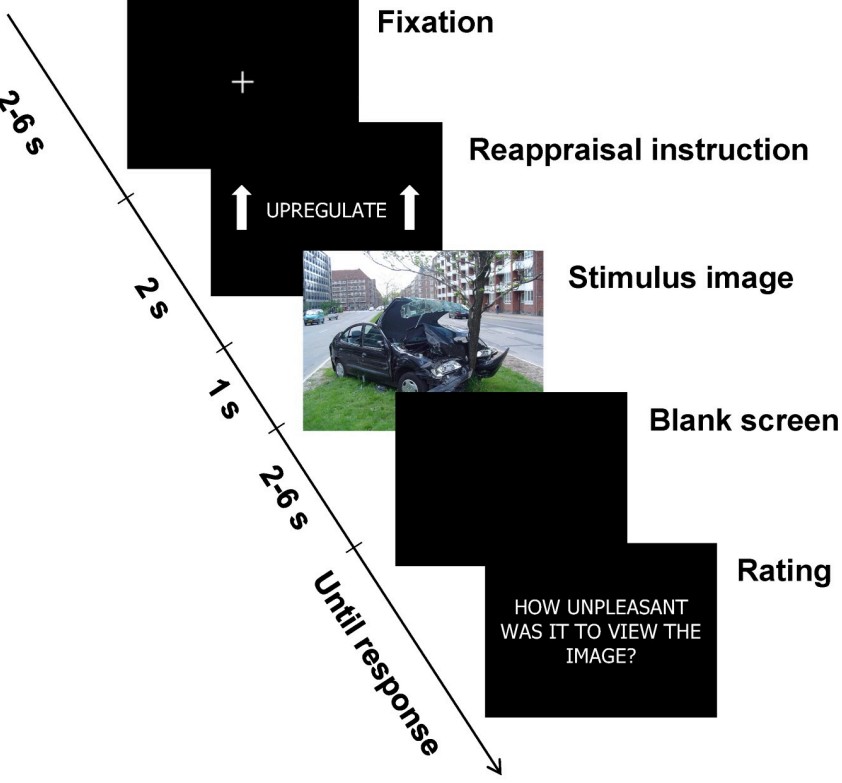

**64 events; 16 of each category:**
**downregulate/upregulate, neutral/negative**

**Fig 1. Experimental paradigm.** The example stimulus image shown here was not part of the stimulus set; the image was created by Wikipedia contributor Thue and released on Wikimedia under a CC0 licence (https://commons. wikimedia.org/wiki/File:Car_crash_1.jpg).

each participant was shown 64 trials, 16 of each category (upregulate/downregulate, neutral/negative). After each stimulus, participants were asked to rate their perceived unpleasantness on a visual analog scale from 0 to 100. Before the experiment, participants underwent a demonstration session, and were then asked to explain the instructions back to the experimenter in order to ensure that the instructions were understood. Stimuli were shown using the Presentation software (Neurobehavioral systems, Inc., Berkeley, CA, USA) on a computer screen. Stimulus presentation code is available at https://doi.org/10.5281/zenodo.31480. For a further verification that emotional regulation took place, we administered a recall test online, where participants were asked to say whether they recalled images from the experiment, as well as valence-matched images from the IAPS that had not been shown in the experiment. The data from the recall test were unfortunately lost when the online test platform upgraded their software.

## Physiological measures

We recorded skin conductance, electromyographic (EMG) activity over the superciliary corrugator muscle, and heart rate, as described in [27].

Briefly, skin conductance responses were measured using Ag/AgCl finger electrodes (TSD203, Biopac Systems, Inc.), connected to a GSR100C amplifier (Biopac Systems, Inc.) with the following acquisitions settings: 5 $\mu\mho$/V, 1 Hz low-pass filter, and direct current. To remove non-physiological noise, data were further filtered in the Acqknowledge software using a low pass filter with a 1 Hz cutoff and 4000 coefficients and converted from direct to alternating current using an 0.05 Hz high pass filter. Responses were averaged over a time window of 2 seconds. The time window for analysis was chosen based on inspection of data.

EMG was measured following established guidelines [30], with 4 mm Ag-AgCl electrodes (EL254S, Biopac Systems, Inc.) connected to EMG100C amplifiers (Biopac Systems, Inc.) with the following acquisition settings: gain 500, low-pass filter 500 Hz, notch filter off, and high-pass filter 10 Hz. Sampling was at 1000 Hz. The signal was further filtered in the Acqknowledge software using a band pass filter of 30 to 300 Hz to remove signal not due to muscle activity. A band stop filter at 49 to 51 Hz was used to filter out line noise. Average rectified EMG signal was determined. Recordings were downsampled to 100 Hz in order to decrease file size, and data were exported as text files. Before analyses, recordings were further downsampled to 10 Hz using a loess curve in R. Responses were averaged over a time window of 2 seconds and log-transformed before statistical analysis, in order to better approximate a normal distribution. The time window for analysis was chosen based on inspection of data.

A 3-lead EKG was acquired using Ag/AgCl electrodes (EL503, Biopac Systems, Inc.) with ECG100 amplifiers (Biopac Systems, Inc.) with the following settings: gain 2000, mode R wave, 35HzLPN on, high-pass filter 0.5 Hz. Sampling was at 1000 Hz. Recordings were downsampled to 100 Hz in order to decrease file size, and data were exported from the Acqknowledge software as text files. Heart rate was derived from raw curves by a peak finding algorithm in R. Estimated heart rate of <40 or >200 beats per minutes was rejected (0.2% of data). For each event, heart rate was normalised to the 2 seconds preceding regulation instruction onset and averaged over a time window from 3 to 5 seconds from regulation instruction onset. The time window for analysis was chosen based on inspection of data.

## Rating scales

The interpersonal reactivity index (IRI) has four subscales which measure different dimensions of trait empathy: empathic concern (EC), perspective taking (PT), personal distress (PD) and fantasy (FS) [31, 32]. The IRI has been validated in a Swedish context [33], although the

four-factor structure could not be replicated. Instead, EC formed one factor and PT, PD and FS together formed another factor. For this reason, we have not analysed differences between IRI subscales.

The state-trait anxiety inventory (STAI) has a state and a trait subscale [34]. We used a non-validated Swedish translation with which we have considerable experience, and which can be found in [35]. The state subscale (S) was administered before the experiment, and then again at the end of the experiment.

The Toronto Alexithymia Scale-20 (TAS-20) measures alexithymia, a construct thought to represent difficulties in identifying and describing one's own emotions. It has three subscales: difficulty identifying feelings, difficulty describing feelings and externally oriented thinking [36]. We analysed only total scores. The scale has been validated in Swedish [37].

The psychopathy personality inventory-revised (PPI-R) assesses psychopathic traits [38, 39]. It contains eight content scales, which have been organized into a two-factor structure, encompassing the factors fearless dominance (FD; reflecting social poise, fearlessness and stress immunity) and self-centred impulsivity (SCI; reflecting impulsivity, irresponsibility and egocentricity). It also contains a subscale particularly reflecting lack of empathy (coldheartedness, C), which typically does not load highly on either factor. The Swedish version of the PPI-R has been validated based partly on the data collected in this study [40]. For more details on the used rating scales, please see [27].

## Analyses and data

Data and analysis code for this paper are openly available at https://doi.org/10.5281/zenodo.3903120. In order to preserve anonymity, participants' ages and educational backgrounds have been omitted from the published dataset. All analyses were made with R [41], using the packages **RCurl** [42] to read data from GitHub, **quantmod** [43] to find EKG R wave peaks, **nlme** [44] to build mixed-effects models, **effects** [45] to get confidence intervals on estimates, and **RColorBrewer** [46] for graphing. Mixed-effects models have been used throughout unless otherwise indicated. Effects were deviation coded, meaning that reported effect sizes refer to the difference from the grand mean. For instance, the effect of upregulation refers to whether participants were instructed to upregulate or downregulate, and the reported effect size is the difference between the upregulate condition and the mean of the upregulate and downregulate conditions.

To investigate interaction effects of self-rated personality traits with stimulus valence and reappraisal instruction, a separate regression model was run with each rating scale, in which interactions between the scale score and valence and reappraisal instruction, respectively, were specified. Scale scores were $z$-transformed to yield standardized regression coefficients.

A threshold of $\alpha < 0.05$ for statistical significance was used because this threshold is conventional in the field.

## Results

### Participants

Thirty-nine participants were randomised. Six did not perform the reappraisal experiment since they reported having previously participated in other experiments involving viewing images likely to be from the same IAPS stimulus set. The final sample included 33 participants. Participant characteristics are shown in Table 1.

**Table 1. Participant characteristics.**

|  | placebo | oxazepam |
| --- | --- | --- |
| n | 13 | 20 |
| age, median (range) | 22 (18-44) | 22.5, 18-41 |
| Interpersonal Reactivity Index—Empathic Concern | 3.77 (0.59) | 3.79 (0.31) |
| Interpersonal Reactivity Index—Perspective Taking | 3.51 (0.37) | 3.41 (0.47) |
| Interpersonal Reactivity Index—Personal Distress | 2.54 (0.39) | 2.39 (0.53) |
| Interpersonal Reactivity Index—Fantasy | 3.37 (0.5) | 3.23 (0.67) |
| State-Trait Anxiety Index—Trait | 40.33 (5.69) | 34.9 (6.21) |
| Toronto Alexithymia Scale-20 | 41.25 (10.32) | 37.5 (8.02) |
| Psychopathy Personality Inventory-Revised—Self-Centred Impulsivity | 160.75 (20.1) | 140.3 (24.74) |
| Psychopathy Personality Inventory-Revised—Fearless Dominance | 125.5 (11.2) | 129.45 (15.25) |
| Psychopathy Personality Inventory-Revised—Coldheartedness | 35.83 (3.07) | 36.05 (4.89) |

Means and standard deviations are given, unless otherwise indicated.

## Unpleasantness ratings

We investigated the interactions between upregulation, negative stimulus valence, and oxazepam treatment on unpleasantness ratings. The three-way interaction was not statistically significant: -2.7 [95% CI -9.1, 3.8], $p = 0.42$ (Fig 2). The two-way interaction between upregulation and negative stimulus valence was statistically significant in the expected direction (12.0 [8.8, 15.2], $p < 0.0001$). The two-way interaction between negative stimulus valence and oxazepam treatment was statistically significant and showed that lower unpleasantness was reported to negative-valence stimuli in the oxazepam group compared to the placebo group (-6.1 [-9.3, -2.7], $p = 0.0002$), consistent with an anxiolytic effect. The two-way interaction between upregulation and oxazepam treatment was not statistically significant: 2.2 [-1.0, 5.4], $p = 0.19$. The main effect of negative stimulus valence was statistically significant: 28.8 [27.2, 30.4], $p < 0.0001$. The main effect of upregulation was statistically significant: 10.2 [8.6, 11.8], $p < 0.0001$, and the main effect of oxazepam treatment was not statistically significant: -1.2 [-9.3, 6.9], $p = 0.76$.

## Skin conductance

Skin conductance was measured as an indicator of autonomic activity. Fig 3 shows time-courses of skin conductance. The time window for signal extraction was chosen based on inspection of time courses.

The three-way interaction between negative stimulus valence, upregulation, and oxazepam treatment was not statistically significant: 0.002 [-0.015, 0.019], $p = 0.79$ (Fig 4). The two-way interaction between negative stimulus valence and upregulation was not statistically significant: -0.006 [-0.014, 0.003], $p = 0.18$. The two-way interaction between negative stimulus valence and oxazepam treatment was not statistically significant: 0.002 [-0.006, 0.011], $p = 0.57$. The two-way interaction between upregulation and oxazepam treatment was not statistically significant: 0.001 [-0.011, 0.013], $p = 0.84$. The main effect of negative stimulus valence was not statistically significant: 0.001 [-0.004, 0.005], $p = 0.78$. The main effect of upregulation was 0.007 [0.003, 0.011], $p = 0.001$, as expected. The main effect of oxazepam was not statistically significant: -0.001 [-0.007, 0.005], $p = 0.71$.

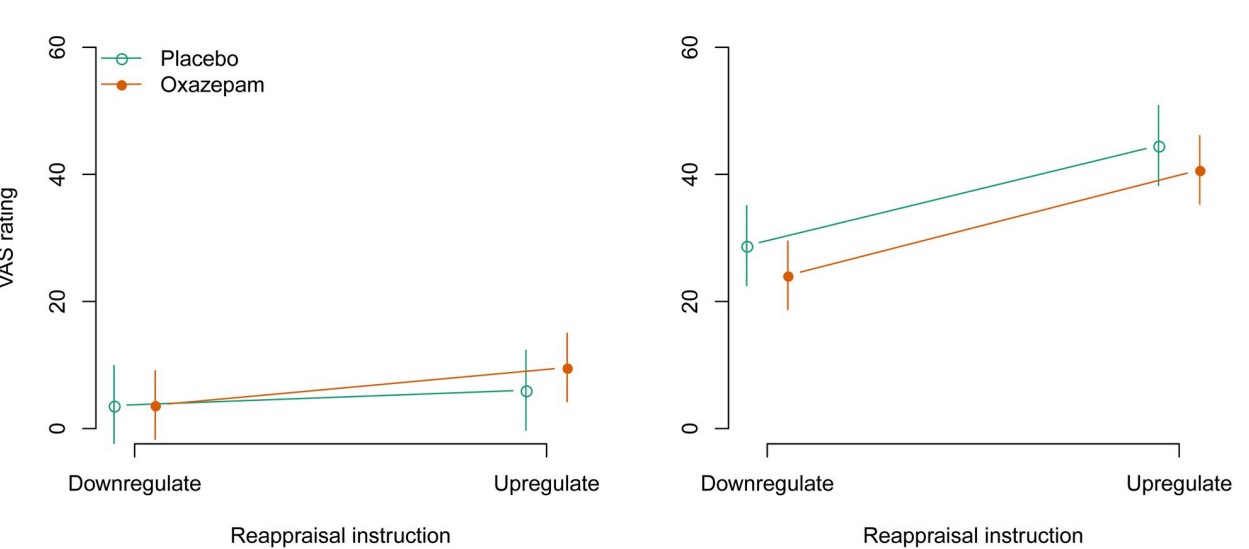

**Fig 2. Rated unpleasantness: Model estimates with 95% confidence intervals.**

## Corrugator activity

Activity of the superciliary corrugator muscle was measured as an indicator of negative emotion. Fig 5 shows time-courses of corrugator EMG activity. The time window for signal extraction was chosen based on inspection of time courses.

The three-way interaction between negative stimulus valence, upregulation, and oxazepam treatment was not statistically significant: -0.031 [-0.201, 0.138], $p = 0.72$ (Fig 6). The two-way

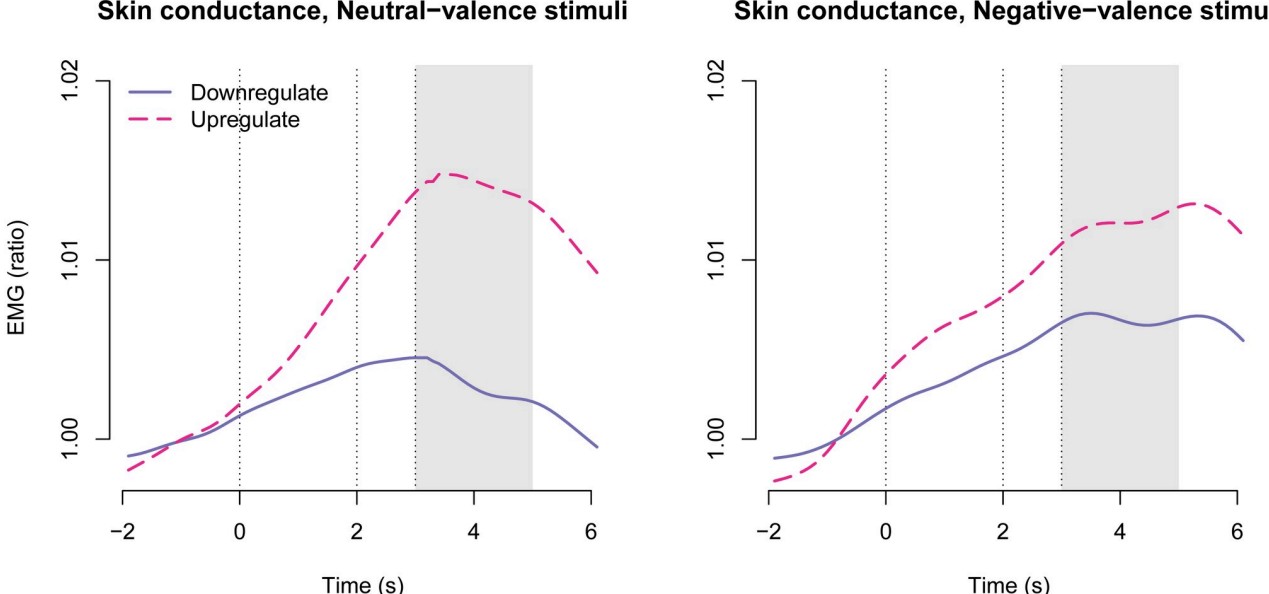

**Fig 3. Skin conductance time-courses across treatment groups.** The first vertical line shows regulation instruction onset; the second vertical line shows stimulus image onset; and the third vertical line shows stimulus image offset. The shaded gray area shows the time window from which responses were averaged for statistical modelling.

## Skin conductance, Neutral−valence stimuli

## Skin conductance, Negative−valence stimuli

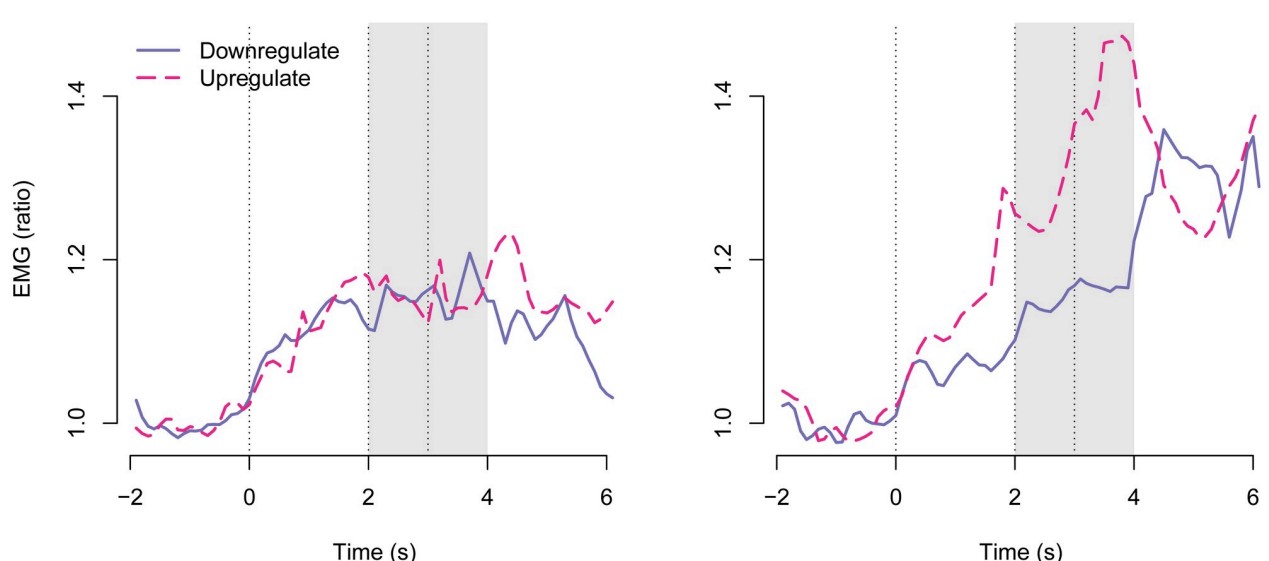

**Fig 4. Skin conductance: Model estimates with 95% confidence intervals.**

interaction between negative valence and upregulation was statistically significant: 0.108 [0.023, 0.192], $p$ = 0.01, as expected, and consistent with unpleasantness ratings. The two-way interaction between negative stimulus valence and oxazepam treatment was not statistically significant: -0.055 [-0.140, 0.030], $p$ = 0.20. The two-way interaction between upregulation and oxazepam treatment was not statistically significant: 0.026 [-0.059, 0.110], $p$ = 0.55. The main

## Corrugator EMG, Neutral−valence stimuli

## Corrugator EMG, Negative−valence stimuli

**Fig 5. Corrugator EMG time-courses across treatment groups.** The first vertical line shows onset of the instruction; the second vertical line shows stimulus image onset; and the third vertical line shows stimulus image offset. The shaded gray area shows the time window from which responses were averaged for statistical modelling.

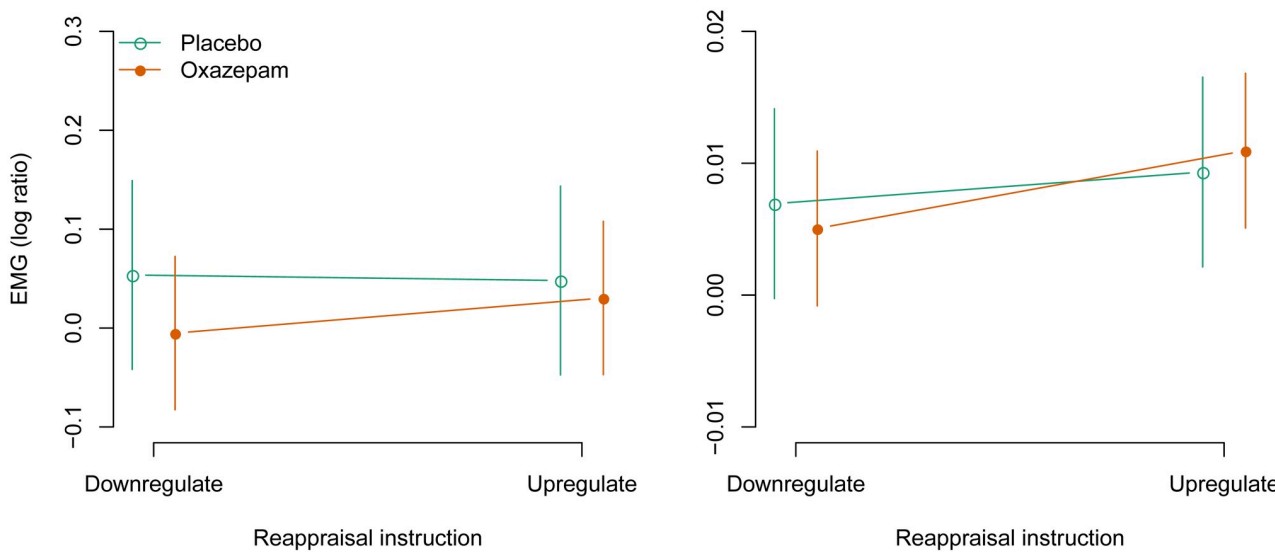

**Fig 6. Corrugator EMG responses: Model estimates with 95% confidence intervals.**

effect of negative stimulus valence was statistically significant: 0.061 [0.019, 0.103], $p$ = 0.005. The main effect of upregulation was statistically significant: 0.069 [0.026, 0.111], $p$ = 0.002. The main effect of oxazepam treatment was not statistically significant: -0.066 [-0.168, 0.037], $p$ = 0.20.

## Heart rate

Heart rate was measured as an indicator of autonomic activity. Fig 7 shows time-courses of heart rate changes, demonstrating deceleration following stimulus presentation. The time window for signal extraction was chosen based on inspection of time courses.

The three-way interaction between negative valence, upregulation, and oxazepam treatment was not statistically significant: -0.013 [-0.037, 0.011], $p$ = 0.29, Fig 8. The two-way interaction between negative valence and upregulation was statistically significant: 0.012 [0.000, 0.024], $p$ = 0.04, as expected. The two-way interaction between negative valence and oxazepam treatment was not statistically significant: -0.001 [-0.013, 0.011], $p$ = 0.90. The two-way interaction between upregulation and oxazepam treatment was not statistically significant: -0.001 [-0.013, 0.010], $p$ = 0.82. The main effect of negative valence was not statistically significant: -0.002 [-0.008, 0.004], $p$ = 0.51. The main effect of upregulation was not statistically significant: 0.003 [-0.003, 0.009], $p$ = 0.39. The main effect of oxazepam treatment was not statistically significant: -0.009 [-0.021, 0.004], $p$ = 0.19, Fig 8.

## Associations between self-rated personality traits to responses to stimuli and instructions to perform reappraisal

We performed exploratory analyses of associations between self-rated personality traits and rated unpleasantness of negative vs neutral images (stimulus valence) as well as instruction to upregulate vs downregulate (instruction). The corresponding statistics can be found in (Fig 9). For stimulus valence, we found that empathy subscales IRI-EC, IRI-PT, and IRI-F were associated with higher rated unpleasantness to negative-valence images, as expected. Conversely,

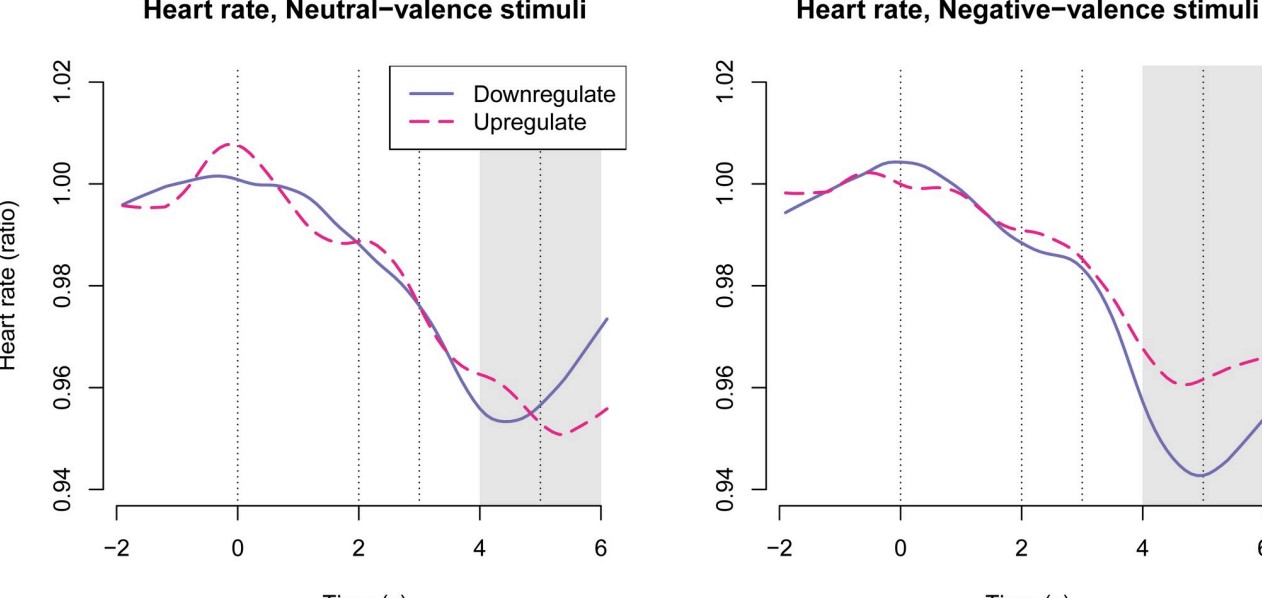

**Fig 7. Heart rate time-courses across treatment groups.** The first vertical line shows onset of the instruction; the second vertical line shows stimulus image onset; and the third vertical line shows stimulus image offset. The shaded gray area shows the time window from which responses were averaged for statistical modelling.

empathy subscale IRI-PD was associated to lower rated unpleasantness, contrary to our expectations. All three subscales of the psychopathy personality inventory-revised (PPI-R) were associated with lower rated unpleasantness to negative-valence images, as expected. For physiological measures, the only notable associations between personality traits and stimulus

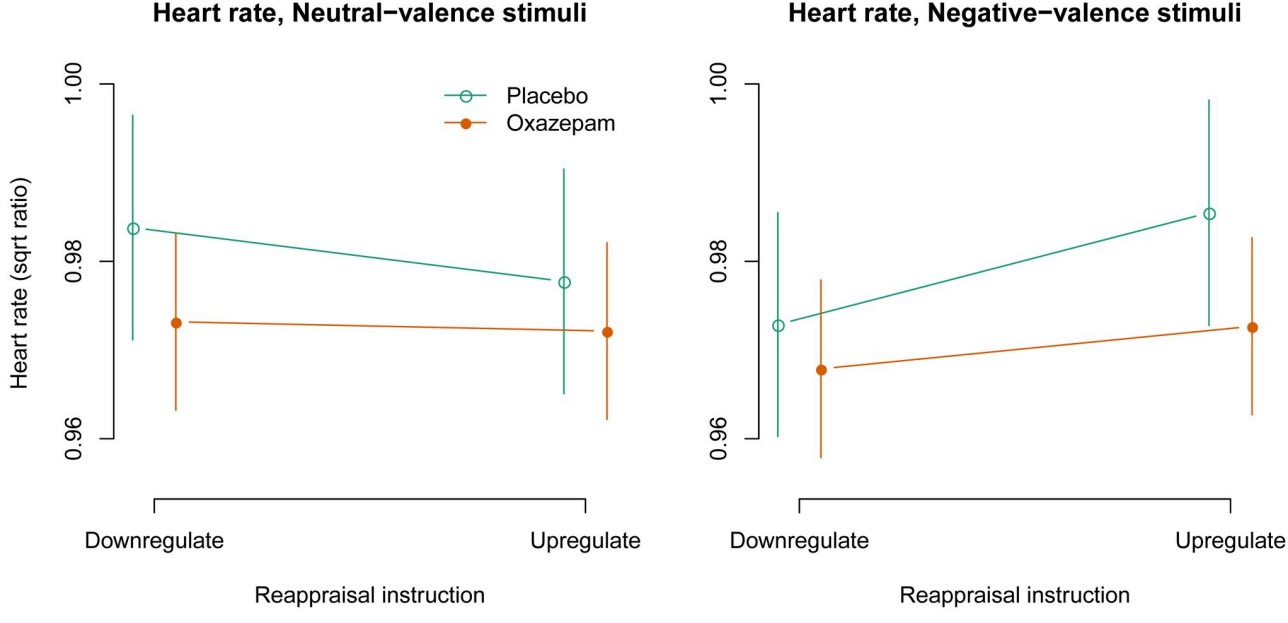

**Fig 8. Heart rate: Model estimates with 95% confidence intervals.**

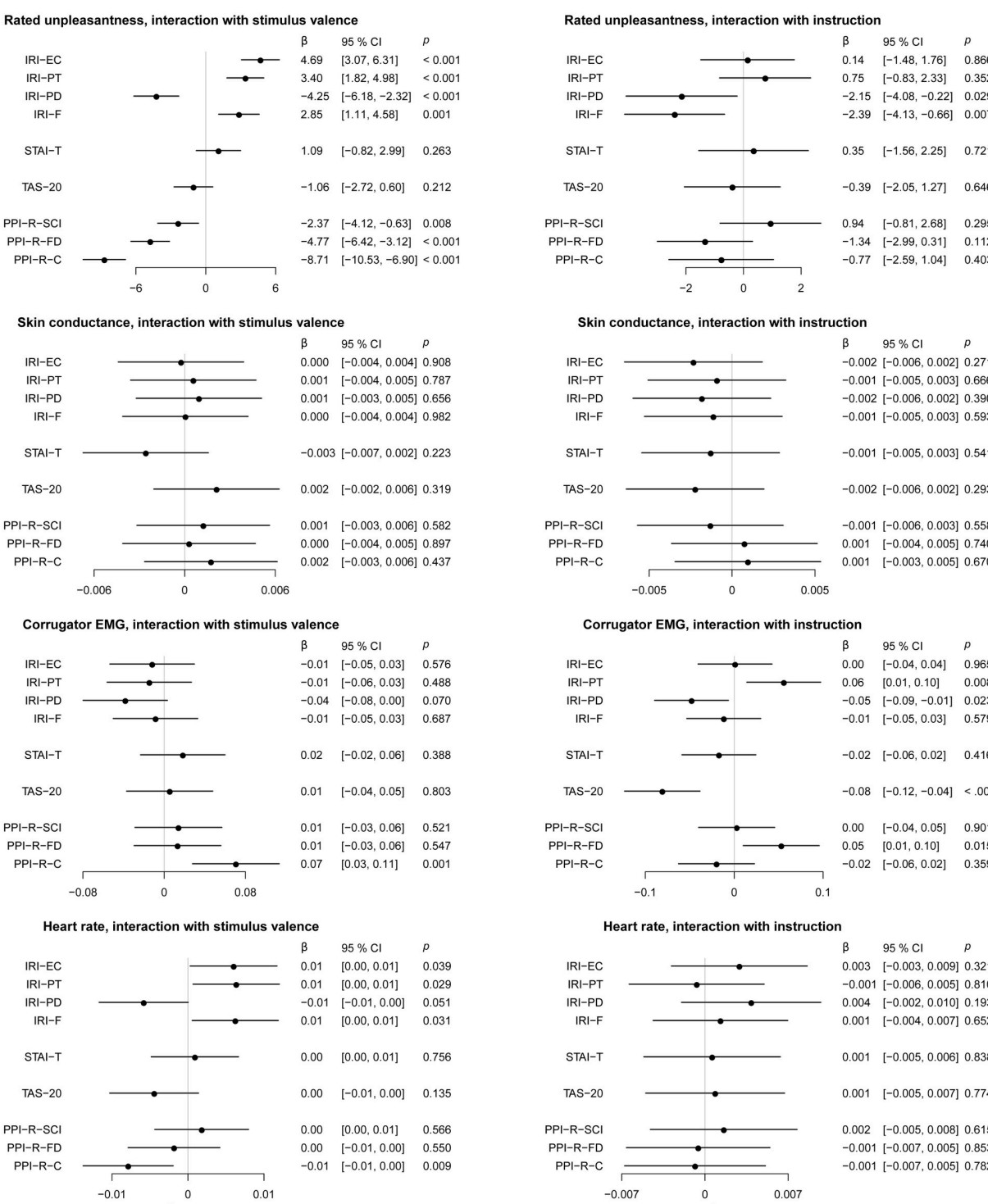

**Fig 9. Interaction effects of self-rated personality traits with stimulus valence and reappraisal instruction.** Effects shown are standardized regression coefficients with 95% confidence intervals.

valence was that PPI-R-C (coldheartedness) was associated with higher corrugator activity and lower heart rate in response to negative images.

For instruction, we found that empathy subscales IRI-PD and IRI-F were associated with lower rated unpleasantness when instructed to upregulate, compared to downregulate. For physiological measures, notable associations between regulation instruction and outcome were seen only for corrugator EMG, where the perspective taking empathy subscale IRI-PT and the Fearless Dominance subscale of the PPI-R (PPI-R-FD) were associated with higher activity in the upregulate condition, whereas the Toronto Alexithymia Scale-20 (TAS-20) and the personal distress empathy subscale IRI-PD were associated with lower activity in the upregulate condition.

## Discussion

In this study, we examined the effect of 25 mg oxazepam on cognitive reappraisal. We found that upregulation of negative-valence images caused increased unpleasantness ratings, corrugator activity, and heart rate compared to downregulation, confirming the validity of the paradigm. Oxazepam caused lower ratings of unpleasantness to negative stimuli, but no interaction between oxazepam and reappraisal was observed.

As we have previously described in a report on the empathy for pain paradigm using the full sample of this experiment, [27], efficacy of drug effects was verified by longer response times and lower rated anxiety after the experiment in the oxazepam group. As noted above, upregulation of negative-valence images caused increased unpleasantness ratings, corrugator activity, and heart rate. Upregulation to both negative- and neutral-valence images also caused increased skin conductance responses. These findings confirm that the experimental paradigm was effective and that the drug reached a biological effect in the oxazepam group, providing an appropriate setting to test the effect of oxazepam on cognitive reappraisal. Ratings of unpleasantness may be affected by demand effects. As argued by Ray et al. [47], it is more convincing to measure both self-report and autonomic indices of emotional responses in reappraisal experiments, since they are sensitive to partly different sets of biases. In particular, autonomic responses are likely to be less affected by demand characteristics. In line with this reasoning, our finding that upregulation by cognitive reappraisal was associated with increased skin conductance responses further supports the validity of the paradigm, and is consistent with earlier findings [10].

Oxazepam caused lower ratings of unpleasantness to negative stimuli, but did not show any noteworthy interaction with cognitive reappraisal on any of our outcomes, contrary to the main hypothesis. These results indicate that 25 mg oxazepam does not have a major effect on cognitive reappraisal. However, the effect on emotional stimuli (without regulation) indicates that oxazepam modulates the affective experience of negative stimuli. This finding is consistent with an earlier finding from our group, where i.v. midazolam reduced perceived unpleasantness of negative-valence IAPS images [48]. It has been suggested that this effect could be caused by a general decrease in anxiety due to inhibited amygdala-dependent emotional processing [12, 25, 26]. Although the same physiological measurements were not used, our results contrast somewhat with previous reports that 20 mg oxazepam did not modulate affective ratings or event-related potentials in response to emotional stimuli [49], and that neither 15 nor 30 mg oxazepam affected the fear-potentiated startle response [50]. Our results are however consistent with a finding that 0.25 and 1 mg alprazolam inhibited startle responses to emotional stimuli [51].

Cognitive reappraisal works through reappraising emotional stimuli in different ways. Functional imaging studies have shown that dorsolateral prefrontal and lateral orbitofrontal

cortices are specifically involved in these processes [7, 9–11]. It has also been shown that these prefrontal processes interact with processing of emotional information in brain regions such as the amygdala [52–54]. Since we showed effects of cognitive reappraisal on affective ratings and physiological measures but no interaction with benzodiazepines our data suggests that these regulatory processes do not interact but work in parallel. Therefore, our data do not corroborate the prediction that top-down regulatory mechanisms are suppressed by benzodiazepines, arising from the disinhibition theory of criminal violent behavior as suggested by [4, 5]. The findings also do not suggest that benzodiazepines, although they have other risks in a clinical setting, contribute to poorer cognitive reappraisal in patients, as might have been expected from the observation that areas in the prefrontal cortex that are involved in emotional regulation also have high concentrations of $GABA_A$ receptors [28].

Associations between on the one hand self-rated empathy, anxiety, alexithymia, psychopathy, and on the other hand cognitive reappraisal, have not been widely investigated. Exploratory analyses in our data showed that self-rated trait empathy measures were associated with stronger responses to negative-valence stimuli (except for personal distress that had the opposite result), whereas self-rated psychopathic traits were associated with weaker responses to negative-valence stimuli. Less consistent effects were observed in relation to instructions. One study has investigated the association of alexithymia to event-related potentials (ERP:s) during cognitive reappraisal, and found that higher alexithymia in a sample of healthy humans was associated with smaller ERP:s [55]. The strongest associations between rating scales and experimental outcomes were observed for self-rated unpleasantness. This may be explained by the similar nature between these self-rated measures, as opposed to physiological measures.

The generalizability of our results is limited by the nature of the sample, consisting of only male participants, most with ongoing or completed university education. This sample is not likely to be representative of benzodiazepine-prescribed patient groups nor recreational users. A further limitation concerns nature of the stimuli, which are a subset of all possible stimuli which could be used to induce emotion. The limited size of the sample, particularly the placebo group, precludes strong conclusions, and as with all randomised experiments, the possibility cannot be ruled out that baseline imbalances may influence the result, though we had no strong *a priori* reason to stratify randomisation to avoid imbalance on some particular variable. Furthermore, the choice of time windows for analyses of physiological signals is based on the observed data, which may introduce bias. Further work should use independent samples to define time windows of interest and to test hypotheses. It is also an open question whether different results would be seen with a higher dose of oxazepam or with another benzodiazepine.

## Conclusion

While 25 mg oxazepam caused lower rated unpleasantness in response to negative valence images, we did not observe an effect of 25 mg oxazepam on cognitive reappraisal.

## Acknowledgments

We are grateful to Jonathan Berrebi for expert technical assistance.

## Author Contributions

**Conceptualization:** Gustav Nilsonne, Sandra Tamm, Armita Golkar, Andreas Olsson, Martin Ingvar, Predrag Petrovic.

**Data curation:** Gustav Nilsonne.

**Formal analysis:** Gustav Nilsonne.

**Funding acquisition:** Gustav Nilsonne, Martin Ingvar.

**Investigation:** Gustav Nilsonne, Sandra Tamm, Armita Golkar, Andreas Olsson, Karolina Sörman, Katarina Howner, Marianne Kristiansson, Martin Ingvar, Predrag Petrovic.

**Methodology:** Gustav Nilsonne, Sandra Tamm, Martin Ingvar.

**Project administration:** Gustav Nilsonne, Martin Ingvar.

**Resources:** Armita Golkar, Martin Ingvar.

**Software:** Armita Golkar.

**Supervision:** Gustav Nilsonne, Martin Ingvar, Predrag Petrovic.

**Visualization:** Gustav Nilsonne.

**Writing – original draft:** Gustav Nilsonne.

**Writing – review & editing:** Gustav Nilsonne, Sandra Tamm, Armita Golkar, Andreas Olsson, Karolina Sörman, Katarina Howner, Marianne Kristiansson, Martin Ingvar, Predrag Petrovic.

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
