## [Decision Letter · Decision Letter 0]

23 Sep 2020

PONE-D-20-19162

Oxazepam and cognitive reappraisal: a randomised experiment

PLOS ONE

Dear Dr. Nilsonne,

Thank you for submitting your manuscript to PLOS ONE. After careful consideration, we feel that it has merit but does not fully meet PLOS ONE’s publication criteria as it currently stands. Therefore, we invite you to submit a revised version of the manuscript that addresses the points raised during the review process.

We look forward to receiving your revised manuscript.

Kind regards,

Alexandra Kavushansky, PhD

Academic Editor

PLOS ONE

Journal Requirements:

Reviewers' comments:

Reviewer's Responses to Questions

**Comments to the Author**

1. Is the manuscript technically sound, and do the data support the conclusions?

Reviewer #1: Yes

Reviewer #2: Partly

2. Has the statistical analysis been performed appropriately and rigorously? 

Reviewer #1: Yes

Reviewer #2: No

3. Have the authors made all data underlying the findings in their manuscript fully available?

Reviewer #1: Yes

Reviewer #2: Yes

4. Is the manuscript presented in an intelligible fashion and written in standard English?

Reviewer #1: Yes

Reviewer #2: Yes

5. Review Comments to the Author

Reviewer #1: The authors implemented most of my previous suggestions in this revision, which has improved the derivation of research questions, description of the study paradigm, and discussions of findings as well as limitations. A few minor issues need to be resolved before publication however.

General: In keeping with general terminology in the manuscript and to avoid confusion, “emotional reappraisal” should be changed to “cognitive reappraisal” throughout the manuscript (see abstract, conclusions, etc.)

Materials and Methods, participants: Since PLoS One is a multidisciplinary journal and not all readers may be familiar with substance interactions, please add a short statement as to why participants could not be habitual consumers of nicotine. Are there interactions of oxazepam to be expected?

Results: The authors adopted the results section according to my previous comments (variable naming), but did not add information as to whether an effect is statistically significant or not. However, I am of the opinion that it would greatly increase comprehensibility on part of the readers to respectively state before reporting effects and p-values whether an effect/interaction was significant or not. Example for unpleasantness ratings: “The three-way interaction was not significant at -2.7….”. This should be changed for all effects and interactions before I recommend publication. Additionally, for skin conductance, main effect of up-regulation and corrugator activity, main effect of negative stimulus valence, the authors should give a short description of the effect instead of simply stating “… as expected”.

Associations between self-rated personality traits (p. 8): Please add that corresponding statistics can be found in Figure 9, as this is not immediately evident.

Discussion:

p.9: the sentence “It has also been shown that these prefrontal processes interact with processing of emotional information in brain regions such as the amygdala” is missing references.

p.9: the sentence “The findings also do not suggest that benzodiazepines […] contribute to poorer emotion regulation performance in patients” needs to be amended.

First, the authors did not directly assess emotion regulation performance, which would require objective analysis of number of reappraisal ideas and quality of ideas (see Papousek et al., 2017; Brain Imaging or Behavior, or Rowlands et al., 2019; Neuropsychological Rehabilitation). Much rather, reappraisal of IAPS pictures assesses individuals’ regulation success or reappraisal effectiveness in terms of differences in emotion intensity and physiological variables between two conditions (regulate up/down, etc.). Next, generalizing this study’s findings to “emotion regulation” in general does not seem justified, since only cognitive reappraisal was investigated. For these reasons, this sentence should read “The findings also do not suggest that benzodiazepines […] contribute to poorer cognitive reappraisal in patients.”

Reviewer #2: Introduction

1. The authors should draw some stronger connections between the GABAa receptors and frontal lobe activity to help solidify the logic behind by cognitive reappraisal would be impacted by benzodiazepines. Within the final paragraph on page one, the authors suggest this connection, but it could use some stronger connections.

2. The authors present aims related to correlations/associations between measures, yet have a set of analyses related to physiological measures involving higher order cause and effect conclusion that can be made. Results should correspond to the predictions made in the aims section.

Method

1. Given that only males were used, the ability to generalize these results is rather limited. Moreover the population of individuals who would take benzodiazepines would likely show the greatest benefit and likely different results for any type of down regulation of emotion as compared to a “normal” sample.

2. What was the average valance and arousal for the sets of IAPS images that were used?

3. How long was each picture shown for? How long did the rating scale appear on the screen for? The authors mention they used 2 second time windows, but it isn’t clear if the stimuli were presented for only 2 seconds or longer.

Results

a. Much of the results section reported results that were not significant. There were a few interaction effects that were typical and expected.

b. The final sample size is reported to be 22, but the table for the placebo condition it appears to say that only 13 participants were in the placebo condition, giving the treatment condition significantly more participants. This could account for a lack of findings in many areas.

c. Corrugator activity shows expected increases with unpleasant stimuli in the upregulation condition, but no impact it appears of the treatment. Moreover, any of the results that are main effects should only be reported and not interpreted if there is a significant interaction within that set of analyses (e.g. the corrugator activity analyses).

d. The majority of the analyses reported appear to have only correlations.

Discussion

The analyses that should have resulted in some of the main conclusions about this study appear to be pushed aside as a mere checks and balances set of analyses. The conclusions that are made about the impact of oxazepam appear to impact the subjective emotional experience but not the physiological experience. That is, the results indeed suggest that the participants may interpret that they should feel a particular way without effectively performing the task.

3. The timecourse of cognitive reappraisal is an important thing to note here. Cognitive reappraisal does not unfold over the course of 2 seconds and the measures that the authors are using rely on autonomic nervous system changes. That is, from an EEG perspective the timecourse of changing brain activity as a result of the cognitive reappraisal task is indeed valid; however, the timecourse to change peripheral physiological responses may differ greatly. This leads back to my question within the method related to how long the stimuli were presented for. It is likely the case that if there was a longer timeframe that the physiological measures were captured there may have been changes late in the stimulus presentation for ECG and skin conductance.

6. PLOS authors have the option to publish the peer review history of their article (what does this mean?). If published, this will include your full peer review and any attached files.

Reviewer #1: No

Reviewer #2: No

---

## [Author Response · Author response to Decision Letter 0]

13 Feb 2021

Response to reviewers

We thank the reviewers and the editor for the valuable comments, which have improved the manuscript. Responses to the points are provided below.

Reviewer #1: The authors implemented most of my previous suggestions in this revision, which has improved the derivation of research questions, description of the study paradigm, and discussions of findings as well as limitations. A few minor issues need to be resolved before publication however.

General: In keeping with general terminology in the manuscript and to avoid confusion, “emotional reappraisal” should be changed to “cognitive reappraisal” throughout the manuscript (see abstract, conclusions, etc.)

This has been changed throughout.

Materials and Methods, participants: Since PLoS One is a multidisciplinary journal and not all readers may be familiar with substance interactions, please add a short statement as to why participants could not be habitual consumers of nicotine. Are there interactions of oxazepam to be expected?

We excluded habitual consumers of nicotine to reduce the risk of abstinence symptoms during the experiment. We have now added this into the methods section (page 3, “Participants”).

Results: The authors adopted the results section according to my previous comments (variable naming), but did not add information as to whether an effect is statistically significant or not. However, I am of the opinion that it would greatly increase comprehensibility on part of the readers to respectively state before reporting effects and p-values whether an effect/interaction was significant or not. Example for unpleasantness ratings: “The three-way interaction was not significant at -2.7….”. This should be changed for all effects and interactions before I recommend publication. Additionally, for skin conductance, main effect of up-regulation and corrugator activity, main effect of negative stimulus valence, the authors should give a short description of the effect instead of simply stating “… as expected”.

This has been changed throughout.

Associations between self-rated personality traits (p. 8): Please add that corresponding statistics can be found in Figure 9, as this is not immediately evident.

This has been added (page 8, “Associations between self-rated personality traits to responses to stimuli and instructions to perform reappraisal”).

Discussion:

p.9: the sentence “It has also been shown that these prefrontal processes interact with processing of emotional information in brain regions such as the amygdala” is missing references.

We have added references as requested.

p.9: the sentence “The findings also do not suggest that benzodiazepines […] contribute to poorer emotion regulation performance in patients” needs to be amended.

First, the authors did not directly assess emotion regulation performance, which would require objective analysis of number of reappraisal ideas and quality of ideas (see Papousek et al., 2017; Brain Imaging or Behavior, or Rowlands et al., 2019; Neuropsychological Rehabilitation). Much rather, reappraisal of IAPS pictures assesses individuals’ regulation success or reappraisal effectiveness in terms of differences in emotion intensity and physiological variables between two conditions (regulate up/down, etc.). Next, generalizing this study’s findings to “emotion regulation” in general does not seem justified, since only cognitive reappraisal was investigated. For these reasons, this sentence should read “The findings also do not suggest that benzodiazepines […] contribute to poorer cognitive reappraisal in patients.”

This has been edited as suggested.

Reviewer #2: Introduction

1. The authors should draw some stronger connections between the GABAa receptors and frontal lobe activity to help solidify the logic behind by cognitive reappraisal would be impacted by benzodiazepines. Within the final paragraph on page one, the authors suggest this connection, but it could use some stronger connections.

We have added text to the introduction and discussion to strenghten this connection in light of PET evidence of high concentrations of GABAA receptors in areas of the prefrontal cortex related to emotional processing (page 2, end of 2nd paragraph, and page 9, 4th paragraph of discussion).

2. The authors present aims related to correlations/associations between measures, yet have a set of analyses related to physiological measures involving higher order cause and effect conclusion that can be made. Results should correspond to the predictions made in the aims section.

We have again reviewed the aims and results for consistency. The main aim was to investigate effects of benzodiazepines, whereas design factors of the experimental paradigm are assumed to have intended effects, e.g. higher unpleasantness to negative compared to neutral images. Validation of the paradigm was not an aim of this study.

Method

1. Given that only males were used, the ability to generalize these results is rather limited. Moreover the population of individuals who would take benzodiazepines would likely show the greatest benefit and likely different results for any type of down regulation of emotion as compared to a “normal” sample.

These important limitations are noted in the discussion (last paragraph).

2. What was the average valance and arousal for the sets of IAPS images that were used?

This has been added under Stimuli and Experimental Paradigm.

3. How long was each picture shown for? How long did the rating scale appear on the screen for? The authors mention they used 2 second time windows, but it isn’t clear if the stimuli were presented for only 2 seconds or longer.

We have added this information in the methods section (“Stimuli and experimental paradigm”), please see also figure 1.

Results

a. Much of the results section reported results that were not significant. There were a few interaction effects that were typical and expected.

We agree with the reviewer’s observation.

b. The final sample size is reported to be 22, but the table for the placebo condition it appears to say that only 13 participants were in the placebo condition, giving the treatment condition significantly more participants. This could account for a lack of findings in many areas.

We agree that this is indeed the case, and we discuss this limitation in the discussion section (last paragraph).

c. Corrugator activity shows expected increases with unpleasant stimuli in the upregulation condition, but no impact it appears of the treatment. Moreover, any of the results that are main effects should only be reported and not interpreted if there is a significant interaction within that set of analyses (e.g. the corrugator activity analyses).

We have removed interpretations of main effects in the presence of interaction effects throughout the results section.

d. The majority of the analyses reported appear to have only correlations.

We agree with the reviewer’s observation.

Discussion

The analyses that should have resulted in some of the main conclusions about this study appear to be pushed aside as a mere checks and balances set of analyses. The conclusions that are made about the impact of oxazepam appear to impact the subjective emotional experience but not the physiological experience. That is, the results indeed suggest that the participants may interpret that they should feel a particular way without effectively performing the task.

We agree that effects of oxazepam are evident in self-rated unpleasantness but not in physiological outcomes. The reviewer suggests that this may be due to demand effects. We discuss this possibility in the manuscript (discussion, 2nd paragraph). 

While we appreciate that the reviewer wishes to further highlight significant effects of the paradigm, this study did not aim to test the validity of the experimental paradigm, which is based on previous research as described in the introduction. 

3. The timecourse of cognitive reappraisal is an important thing to note here. Cognitive reappraisal does not unfold over the course of 2 seconds and the measures that the authors are using rely on autonomic nervous system changes. That is, from an EEG perspective the timecourse of changing brain activity as a result of the cognitive reappraisal task is indeed valid; however, the timecourse to change peripheral physiological responses may differ greatly. This leads back to my question within the method related to how long the stimuli were presented for. It is likely the case that if there was a longer timeframe that the physiological measures were captured there may have been changes late in the stimulus presentation for ECG and skin conductance.

Effects were measured up to 6 seconds; please see figures 3, 5, and 7.

---

## [Decision Letter · Decision Letter 1]

11 Mar 2021

Oxazepam and cognitive reappraisal: a randomised experiment

PONE-D-20-19162R1

Dear Dr. Nilsonne,

We’re pleased to inform you that your manuscript has been judged scientifically suitable for publication and will be formally accepted for publication once it meets all outstanding technical requirements.

Kind regards,

Alexandra Kavushansky, PhD

Academic Editor

PLOS ONE

Additional Editor Comments (optional):

Reviewers' comments:

Reviewer's Responses to Questions

**Comments to the Author**

1. If the authors have adequately addressed your comments raised in a previous round of review and you feel that this manuscript is now acceptable for publication, you may indicate that here to bypass the “Comments to the Author” section, enter your conflict of interest statement in the “Confidential to Editor” section, and submit your "Accept" recommendation.

Reviewer #1: All comments have been addressed

2. Is the manuscript technically sound, and do the data support the conclusions?

Reviewer #1: Yes

3. Has the statistical analysis been performed appropriately and rigorously? 

Reviewer #1: Yes

4. Have the authors made all data underlying the findings in their manuscript fully available?

Reviewer #1: Yes

5. Is the manuscript presented in an intelligible fashion and written in standard English?

Reviewer #1: Yes

6. Review Comments to the Author

Reviewer #1: The authors have addressed all my comments to my satisfaction. The manuscript has improved significantly from previous submissions and is now suitable to be published in PLOS One.

7. PLOS authors have the option to publish the peer review history of their article (what does this mean?). If published, this will include your full peer review and any attached files.

Reviewer #1: No

---

## [Editor Report · Acceptance letter]

12 Apr 2021

PONE-D-20-19162R1 

Oxazepam and cognitive reappraisal: a randomised experiment 

Dear Dr. Nilsonne:

I'm pleased to inform you that your manuscript has been deemed suitable for publication in PLOS ONE. Congratulations! Your manuscript is now with our production department. 

Kind regards, 

on behalf of

Dr. Alexandra Kavushansky 

Academic Editor

PLOS ONE